# Circulating Proteins as Diagnostic Markers in Gastric Cancer

**DOI:** 10.3390/ijms242316931

**Published:** 2023-11-29

**Authors:** Ombretta Repetto, Roberto Vettori, Agostino Steffan, Renato Cannizzaro, Valli De Re

**Affiliations:** 1Facility of Bio-Proteomics, Immunopathology and Cancer Biomarkers, Centro di Riferimento Oncologico di Aviano (CRO), National Cancer Institute, IRCCS, 33081 Aviano, Italy; 2Immunopathology and Cancer Biomarkers, Centro di Riferimento Oncologico di Aviano (CRO), National Cancer Institute, IRCCS, 33081 Aviano, Italy; rvettori@cro.it (R.V.); asteffan@cro.it (A.S.); 3Oncological Gastroenterology, Centro di Riferimento Oncologico di Aviano (CRO), National Cancer Institute, IRCCS, 33081 Aviano, Italy; rcannizzaro@cro.it; 4Department of Medical, Surgical and Health Sciences, University of Trieste, 34127 Trieste, Italy

**Keywords:** biomarkers, gastric cancer, serum proteins, plasma proteins, circulating biomarkers, proteomics, liquid biopsy, saliva proteins, urine proteins, exosomes

## Abstract

Gastric cancer (GC) is a highly malignant disease affecting humans worldwide and has a poor prognosis. Most GC cases are detected at advanced stages due to the cancer lacking early detectable symptoms. Therefore, there is great interest in improving early diagnosis by implementing targeted prevention strategies. Markers are necessary for early detection and to guide clinicians to the best personalized treatment. The current semi-invasive endoscopic methods to detect GC are invasive, costly, and time-consuming. Recent advances in proteomics technologies have enabled the screening of many samples and the detection of novel biomarkers and disease-related signature signaling networks. These biomarkers include circulating proteins from different fluids (e.g., plasma, serum, urine, and saliva) and extracellular vesicles. We review relevant published studies on circulating protein biomarkers in GC and detail their application as potential biomarkers for GC diagnosis. Identifying highly sensitive and highly specific diagnostic markers for GC may improve patient survival rates and contribute to advancing precision/personalized medicine.

## 1. Gastric Cancer

Gastric cancer (GC) is the fifth most common cancer and the fourth leading cause of cancer deaths in both genders combined worldwide according to the newest data published by the World Health Organization (WHO) in 2020 [1]. In the early stages, this disease is usually asymptomatic or without specific symptoms, with the diagnostic procedure being unnecessarily extended. Around 80% of GC diagnoses are made in the advanced stages when symptoms such as abdominal pain or weight loss are present, and there are limited possibilities for treatment [2]. The major risk factors for GC are *Helicobacter pylori* (*H. pylori*) and Epstein–Barr virus infection, chronic inflammatory processes, excessive consumption of alcohol and meat, smoking, high salt intake, obesity, low consumption of fruits and vegetables, and a family history of blood group A or GC [3]. A family history of GC is reported in ~10–15% of GC cases [4]. To date, despite lifestyle and prevention strategies for GC to reduce patients’ exposure to risk factors, along with the screening and detection of precancerous/early lesions, GC outcomes are still poor, and the five-year survival rate for GC patients is under 30% [5,6].

The Laurén histopathological classification separates gastric adenocarcinomas into two major histological subtypes: diffuse and intestinal [7]. These subtypes exhibit distinct morphologic appearance, pathogenesis, and genetic profiles. The diffuse type (undifferentiated) is characterized by poorly cohesive tumor cells growing as isolated or small cell clusters, and it is more frequently reported in young women and subjects with cancer-positive histories. The intestinal type (well differentiated) composed of cohesive tumor cells mostly organized in tubular, glandular, or papillary structures is primarily associated with chronic atrophic gastritis and develops typically in older patients, men, and persons from high-risk countries [8]. The loss of the cell adhesion E-cadherin protein expression from CDH1 gene alterations is the primary carcinogenetic event in hereditary diffuse GC. This loss activates oncogenic signaling pathways and promotes cancer cell growth and dissemination [9]. On the other hand, intestinal GC is believed to develop via a multistep process starting from chronic gastritis primarily triggered by *H. pylori* and progressing from atrophy, intestinal metaplasia, and dysplasia/intraepithelial neoplasia to early carcinoma. Molecular markers have been reported for intestinal metaplasia (MUC2, MUC5AC, and MUC6) [10], gastric dysplasia (mucins phenotype) [11], and early GC (GATA6, TP53^mut/LOH^, and MUC6) [12]. Moreover, a new classification subdivides GC disease into four subclasses, depending on the presence of (i) Epstein–Barr virus infection, (ii) microsatellite instability, (iii) genomic stability, and (iv) chromosomal instability [13].

So-called early GC is characterized by a limited local cancer progression to the mucosa and submucosa, with/without metastatic lymph node involvement, and commonly has a favorable prognosis in contrast to late-stage (or advanced) GC. Since early GC is often asymptomatic, a cancer diagnostic delay usually occurs when the pathological scenario has become advanced. Therefore, detecting early GC lesions is still a considerable challenge for offering minimally invasive treatments [14].

Indeed, the current techniques for GC diagnosis are mostly invasive, such as pathological examination after biopsy via gastroscopy [15]. Although this is the gold standard for GC diagnosis, upper endoscopy can cause pain and discomfort in patients. In clinical practice, less invasive support should include various imaging techniques, including computed tomography and magnetic resonance imaging, positron emission tomography, and endoscopic ultrasound scanning [16].

In this context, laboratory assays may also offer less expensive and non-invasive solutions. Accordingly, over the last few decades, more studies have been performed to investigate the non-invasive and effective biomarkers for GC diagnosis and identify effective biomarkers for the early detection of GC. Currently, traditional serum cancer markers, including CEA, CA19-9, and CA125, are mostly used in the screening and surveillance (therapy monitoring) of GC rather than for early detection [17] because of their relatively low sensitivity and accuracy [18]. In addition, other extensively studied serum biomarkers (e.g., pepsinogens and anti-*H. pylori* IgG antibodies) may detect gastric precancerous lesions, though with modest sensitivity for cancer. Therefore, there is still a lack of ideal serum/plasma GC screening methodologies, and novel biomarkers must be explored. New attractive methods have detected the following as targets, yielding promising results: cancer-specific methylation patterns [19], circulating tumor cells [20], extracellular vesicles [21], mutations in circulating tumor DNA, cell-free DNA [22], cell-free RNA, and miRNA panels [23].

At present, the detection of highly sensitive and specific circulating protein biomarkers, single or combined, is very attractive. Cancer liquid biopsy plays a central promising role in precision medicine and cancer management, including cancer screening for early detection [24]. Through minimally invasive procedures, it is possible to obtain samples for cancer detection and target both cell-free circulating proteins and those extracted from cell compartments or sub-cellular structures [25]. Because of the great complexity of proteomes in liquid biopsy samples, there are several in-progress efforts to overcome the limitations of proteomic technologies compared to their counterpart: high-plex genomic technologies. Cancer proteomics takes advantage of the innovative developments of robust, high-throughput, standardized, and affordable analytical tools in high-plex formats capable of measuring at least hundreds of proteins simultaneously, ranging from the two major traditional techniques (those based on antibody/antigen array and mass spectrometry) to innovative ones (those based on aptamer, proximity extension assay, and reverse-phase protein arrays) [26].

Circulating proteins, including tumor-secreted proteins via various secretory pathways (the so-called “cancer cell secretome”) or immune system inducers/effectors, are involved in various biological functions. Globally, they may play an important role in cancer development and progression and are thus considered an important potential source of “sentinel molecules”. In particular, the cancer secretome consists of proteins (e.g., extracellular matrix proteins, enzymes, growth factors, inflammatory cytokines, exosomes, and microvesicles) secreted or released by cancer or cancer-related cells or different types of cells, which are a part of the dynamic interactions within the highly complex local tumor milieu [27,28,29]. Proteins released into the extracellular compartment or bodily fluids may activate finely orchestrated signaling pathways driving tumor microenvironment remodeling, tumor growth, and diffusion [30].

Circulating proteins are thus a major origin of cancer biomarkers. The blood proteome is composed of tissue proteins and blood-resident proteins [31]. Blood protein abundance variations may reflect the general health condition of patients and can be analyzed to monitor disease progression [32]. Malignant cells develop many properties for their progression and metastasis, such as the manipulation of the immune system checkpoints or the induction of growth, neoangiogenesis, and invasion [33]. Circulating proteins, when among other molecules, can actively contribute to these new cell behaviors.

Most cancer-secreted proteins participate in different biological and physiological events, such as immune response, inflammation, and cell–cell molecular dialogue. The cancer secretome can be measured in blood and other human fluids. Those secreted proteins are putative cancer markers and are thus easier to access than proteins within tumor tissue. The secretome has been only partly deciphered in GC [34,35].

Today, non-invasive approaches for preparing human samples for biomarker discovery are being widely established. Blood protein analysis is becoming a routine and frequent method. The identification of quantitative changes in circulating proteins has been performed by clinical laboratories for a long time. Recent advances in the development of new technologies for protein analysis, including enzyme-linked immunosorbent assay (ELISA), mass spectrometry (MS), or antibody array, have increased the capacity and specificity of these assays, enabling the detection of hundreds or thousands of proteins, including the low-abundant ones. Proteomics investigations of circulating biomarkers in GC combine both identification and quantification, and they are both “targeted”, particularly based on ELISA immunoassay panels on a limited number of target analytes, and “untargeted”, preferentially based on MS methods enabling large-scale workflows. Overall, targeted and untargeted blood proteomics appear to be a favorable approach to discovering new biomarkers, taking advantage of high-throughput technologies [36]. Different technologies and refined pipelines are currently available in research settings for biomarker discovery and protein profiling in various body fluids that are alternatives to blood (recently reviewed by Dayon et al. [37]), such as saliva [38], gastric juice [39], ascites [40], and urine [41]. The traditional ELISA technique, based on an antigen–antibody reaction without any complex sample pre-treatment, exhibits several advantages: (i) simplicity, (ii) high specificity and sensitivity, (iii) high efficiency, (iv) relatively short turnaround time, (v) low sample cost, and (vi) automation. However, it also exhibits some disadvantages: (i) lack of multiplexing since only one single analyte can be analyzed, (ii) high risk of false positive or negative results, and (iii) possible antibody instability. At present, new multiplex ELISA are available and allow one, together with the Luminex-based technology, to overcome one limit of the conventional one-plex ELISA. During the last few years, rapid developments in MS-based proteomics with a particular emphasis on its application to clinics have been extensively obtained, taking advantages of (i) new sample preparation procedures, enabling simplification of the highly complex nature of body fluids; (ii) developments in MS equipment and configuration with improved sensitivity, resolution, and specificity; and (iii) new software and algorithms to analyze and statistically evaluate MS-based proteomics data (reviewed by Birhanu et al. [42]). Differently from immunometric assays like ELISA, which are widely used in clinics, MS-based techniques still lack automation in sample preparation and interfaces between the instrument and laboratory information system and result transmission. Moreover, MS-based proteomics needs skilled personnel and has high capital costs. However, compared to the most traditional targeted ELISA, a typical MS-based proteomics workflow (“targeted” or “untargeted”) allows for the identification of hundreds of putative protein markers.

In this review, we present updated information about circulating proteins in the blood (Table 1) and other body fluids (Table 2) and about extracellular exosomes, which may represent predictive biomarkers of GC (Figure 1). Moreover, we update information concerning glycosylation and its dysregulation in GC as a potential diagnostic protein hallmark.

## 2. Circulating Protein Biomarkers: An Update over the Past 10 Years

Over the last few decades, the characterization of circulating proteins has shown consistent advantages from the continuing progress achieved via proteomics. Globally, several analytical platforms for proteomics have allowed us to identify the entire set of human proteins and uncover qualitative and quantitative variations of numerous proteins upon different stimuli. Typically, an ideal protein biomarker should be a molecule whose level significantly changes in the presence of a disease (either as an increase or as a decrease) so that its abundance can predict the occurrence of the disease itself. Moreover, the differential content of the protein marker should relate to some clinical parameters (i.e., cancer stage, size, invasion depth, degree of differentiation). An ideal biomarker should be quantifiable as a continuous variable that is based on confident reference intervals for clinical decisions.

Biomarker development is typically divided into three phases: the biomarker discovery phase, where biomarkers are identified; this should be followed by a verification phase to confirm the identity and differential expression of the candidates and a validation phase to validate biomarker performance in larger cohorts, leading to robust markers [97]. Normally, the number of samples increases from the discovery to verification cohorts, while the number of putative biomarkers to be validated decreases. Since only a small number of patient samples is often available, many proteomics analyses are still performed on small cohorts. Therefore, the main ambitious goal is still the identification of reliable markers, avoiding false positives due to chance correlations, together with an exhaustive detection of all candidate markers, to get a better insight into the molecular scenario of the disease being investigated.

### 2.1. Blood-Based Circulating Biomarkers

Among the different biological fluids, blood represents the preferential sample for screening tests, including those measuring proteins. Protein markers may accumulate in tissue(s) and body fluids, such as blood, along with cancer development, and variations in the protein profiles/distribution in tissues and the blood can be investigated through qualitative/quantitative proteomics. Abundances of most blood proteins may reach very low concentrations, thus necessitating the use of highly sensitive techniques for quantification [98]. Considerable efforts have been made to characterize the protein content in both serum and plasma in-depth, taking advantage of the rapid advances in sample preparation (i.e., the removal of highly abundant proteins) [99], protein/peptide separation (particularly chromatography) [100], mass spectrometers, and bioinformatics [42,101,102]. In particular, clinically relevant cancer biomarkers in the blood have been investigated using “in gel” or “gel-off” proteomics [103,104,105] using “untargeted” or “targeted” approaches on “singleplex” or “multiplex” panels [106].

Over the past few decades, in GC biomarker discovery, both plasma and serum have been extensively analyzed in terms of proteins, despite their highly complex nature, with an extremely large dynamic range of protein concentration requiring high-resolution separation techniques and enrichment steps [100]. In particular, liquid chromatography–mass spectrometry (LC/MS) is a powerful analytical approach to obtain high-resolution peptide spectra facilitating the identification of cancer-related biomarkers [106]. The quantitation strategies mostly adopted in clinical studies are label-based (e.g., the isobaric “Tandem Mass Tag, TMT” based reporter methodology) or label-free (e.g., the so-called “label free quantification, LFQ” approach), and they allow for the quantitative and qualitative investigations of proteins in a biological matrix [107]. In particular, the TMT methodology can simultaneously identify and quantify target proteins with high-order multiplexing (up to 18 samples) with the lowest system error and high sensitivity [108]. The LFQ approach does not require any labeling, and protein abundance comparisons are based on the relative intensities of extracted ion chromatograms from enzymatic digested peptides [109]. Both approaches have been adopted in workflows to discover blood diagnostic biomarkers in GC (e.g., label-based [44] and label-free [49]). Among other technologies adopted in cancer to screen for putative biomarkers, immunoassays are targeted biomedical techniques commonly used to detect the expression of an antibody or antigen in a test sample, and they include both singleplex (the most known and used is ELISA), where a single analyte is analyzed, and multiplex, where more analytes are quantified, such the xMAP-based technology of Luminex and the immunoblot-based protein pathway array method [110]. Both single and multiplex targeted approaches have been used in proteomics analyses to discover diagnostic biomarkers in the blood of GC patients (e.g., ELISA-based single-plex [43] and Luminex-based multiplex [47]).

Over the past few decades, several studies on blood-based protein biomarker discovery for GC diagnosis, even if they succeeded in proposing some candidates (e.g., AFP, CEA, CA19-9, CA72-4, CA125), evidenced their low specificity and sensitivity [17,70,111,112,113,114], thus limiting their clinical application.

Therefore, at present, there is still a lack of ideal plasma or serum GC diagnostic methods, and new biomarkers must be explored. Ongoing studies are focusing on identifying novel biomarkers for more efficient GC (early) diagnosis. Table 1 details some works applied to the discovery of blood-based (plasma/serum) biomarkers for GC diagnosis over the last 10 years. By adopting different proteomic approaches (e.g., ELISA and LC-MS) on cohorts of patients heterogeneous for both clinical characteristics (e.g., histology, stage) and sampling sizes, these works led to several putative biomarkers, thus confirming the high difficulty in discovering universal diagnostic markers, either as a single protein or as a panel of combined proteins, because of the highly complex biology of the disease. In particular, the ELISA-based technique targeted on analytes related to the immune system (sHLA-G [43,50], PD-1, and PD-L1 [45,61], inflammation (TNF-α [51,55], IL-6 [55,56,70]), ITIH4 [63,69,83], or digestion (PGI, PGII [53,55,81], and GKN1 [66]) is still the most used approach to investigate protein biomarkers in the blood of GC patients, as it has been over the last 10 years. 

Together, the proposed protein biomarkers for GC diagnosis cover a wide range of biological processes (Figure 2), each of them being already characterized in GC pathology, ranging from signal transduction (EGFR/HER2, p53, PI3K, immune checkpoint pathways, and cell adhesion signaling molecules) [115] to inflammatory/immune response [116], the negative regulation of apoptotic processes [117], the positive regulation of cell proliferation, angiogenesis [118], and acute phase response [119].

A single protein failed to behave as an adequate diagnostic marker, which is consistent with the genetic heterogeneity of GC malignancy. Recently, very few targeted studies have focused on the characterization of only one blood protein as a putative diagnostic marker of GC (i.e., plasma sHLA-G [43,50], plasma DEK [48], and serum IGF-1), with most targeted works investigating more combined proteins (i.e., PD1 and PDL1 [45,61], PGI and PGII [53,55], cytokines, and, particularly, IL-6 [55,56,70] and TNF-α [51,55]. In this context, most recent studies have highlighted consistent improvements in specificity/sensitivity levels through the combination of more proteins into one panel test, gaining a level of diagnostic power that cannot be achieved by testing a single protein alone. Overall, independently of the adopted proteomics approach, analyses investigating the same target showed concordant results: plasma HLA-G levels were higher in GC patients compared with those of individuals affected by benign gastric disease or healthy subjects [43,50], serum IL-6 was more abundant in GC patients [55,56,70], and PD-1 content was lower in GC compared with controls [45,61]. However, it should be noted that attempts to relate levels of the proposed diagnostic protein marker(s) to cancer clinical characteristics mostly failed: for instance, sHLA-G was not related to GC stages [43]. Interestingly, a protein signature composed of 19 proteins succeeded in being related to the TNM I-II stage (sensitivity = 89%; specificity = 100%; AUC = 0.99) and high microsatellite instability (91%, 98%, and 0.99) [65].

Apart from the intraindividual heterogeneity of a specific candidate protein biomarker, a certain level of abundance variation may come from the intrinsic GC tumor genetic heterogeneity: investigations specifically relating, for instance, protein abundance with the four molecular GC subclasses [13], to our knowledge, still need to be performed and may represent an opportunity to improve therapeutic outcomes through better early diagnosis.

Untargeted approaches have allowed for the findings of proteins not previously reported as related to GC. For instance, using TMT labeling quantitative proteomics with LC-MS/MS, Zhou et al. [44] compared sera protein profiles from a cohort of GC patients (*n* = 15) with those from a cohort of healthy individuals (*n* = 15) and identified a total of 11 differentially abundant proteins (7 increased: matrix Gla protein, proline–serine–threonine phosphatase-interacting protein 2, neuroblastoma suppressor of tumorigenicity 1, leukocyte immunoglobulin-like receptor subfamily A member 2, folate receptor β, and out at first protein homolog and proprotein convertase subtilisin/kexin type 9; 4 decreased: superoxide dismutase [Cu-Zn], ankyrin-1, ubiquitin-40S ribosomal protein S27a, and uncharacterized protein), which were used to build a logistic regression model more successful in discriminating early GC (sensitivity = 66.7% and specificity = 86.7%) than any individual proteins. In a cohort of 219 patients infected or not by *H. pylori* and suffering from mild to advanced gastritis and ulcers, considered as pre-malignant conditions, and early to advanced GC, using label-free comparative proteomics with LC-MS/MS, Aziz et al. [52] found two serum protein marker panels associated with early or advanced GC independent of *H. pylori* infection, with 29 (i.e., integrin-6 and glutathione peroxidize) and 10 (i.e., CRP, protein S100A9, and kallistatin) proteins, respectively, which were proposed for the further development of multi-protein assays for GC serum diagnostics.

### 2.2. Non-Blood-Based Circulating Biomarkers

At present, although both plasma and serum have proved to be good biological sources for promising new and non-invasive disease biomarkers, their clinical use is still limited by their complex proteomes, which need labor-intensive sample preparation. Therefore, in addition to plasma and serum, other matrices provided the basis to explore cancer and discover new putative diagnostic biomarkers, including ascitic fluid [120,121], gastric juice [39], saliva [91], and urine [122,123]. Patient-based fluid proteomics is a promising approach to search for cancer biomarkers. The proteomes of 10 body fluids, including ascites, plasma, saliva, serum, and urine, have been recently characterized into 3396 nonredundant identified proteins, of which around 10% were shared with common functions in focal adhesion and complement/coagulation cascades [124].

Ascitic fluid is a valuable source of cancer biomarkers since it contains many secreted/shed proteins from cancerous cells. The development of malignant ascites mostly develops in GC advanced stages [125] and is associated with a very poor prognosis, determining if it resulted from peritoneal seeding being critical regarding the diagnosis [126]. Thus, targeted proteomics of ascites on known sentinel proteins may help to gain better insights into the pathophysiology of peritoneal seeding and guide the development of alternative diagnostic methods. In advanced GC (*n* = 85), using an untargeted proteomic approach based on LC-MS/MS, Jin et al. [88] succeeded in identifying protein profiles associated with malignant versus benign ascites and found that two proteins (progastriscin or pepsinogen C, PGC; periostin, POSTN) may be candidate biomarkers of advanced disease.

Gastric juice is another promising source for biomarker discovery, as recently reported by Felipez et al. [39]: although it represents a gastroscopy waste product, its unique characteristic is that it is an exclusive stomach fluid, i.e., it can be considered a “liquid biopsy” characterized by disease-enriched biomarkers and, by containing stomach lining secretions, reflects variations depending on the GC developmental stage. At present, adopting different approaches, analyses performed on GC gastric juice allow us to identify different diagnostic biomarkers of GC: the increase in synuclein-gamma (SNCG) observed via ELISA in serum [59]; the increase in elastase 3A (Ela3A) and a decrease in pepsin A (PepA), gastric lipase (GastL), gastricsin, and Cystatin D (CystD) found via iTRAQ labeling and LC-MS/MS [89]; and S100 calcium-binding protein A9 (S100A9) with α-1-antitrypsin (AAT) analyzed via two-dimensional electrophoresis, followed by mass spectrometry [90].

The choice of saliva as a biomarker source is an alternative attractive approach for GC screening because it is easily accessible, its production via salivary glands may be induced by molecules released from cancer, and its proteins may reflect a myriad of functions altered in the presence of disease [127]. Over 1000 unique human saliva proteins identified using high-throughput proteomics techniques represent a growing database publicly available at https://salivaryproteome.org, accessed on 8 March 2016. In two cohorts of GC patients (discovery and validation), by adopting in-gel and gel-off salivary proteomics, Xiao et al. found that the combination of three proteins (cystatin B, CSTB; triosephosphate isomerase, TPI1; deleted in malignant brain tumors 1 protein, DMBT1) had abundances that were lower in GC saliva, differentiating GC from healthy control patients (*p* < 0.05; sensitivity = 85%; specificity = 80%; accuracy = 0.93) [100]. Although this study, as evidenced by the authors, demonstrated the great potential of salivary biomarkers for the non-invasive detection of GC, to our knowledge, this is the only investigation into salivary proteins in GC over the last 10 years.

Urine, as a minimally invasive source, is advantageous for disease marker discovery, owing to its easy accessibility, high thermodynamic stability, and relatively unlimited sampling volumes. Urine is a promising medium for clinical research because of its less complex protein content than plasma/serum [128]. A comprehensive study on the human urinary proteome reported 1823 proteins in normal human urine [129]. In recent years, an increasing number of studies have adopted different urinary proteomics workflows, e.g., LC-MS/MS, to discover GC diagnostic markers: in cohorts of GC patients differing in number and clinics, the increase in sortilin 1 (SORT1), vitronectin (VTN) [92], annexin A11 (ANXA11), cell division control protein 42 homolog (CDC42), NSF attachment protein α (NAPA), solute carrier family 25 member 4 (SLC25A4) [93], disintegrin and metalloproteinase domain-containing protein 12 (ADAM12), with either Trefoil Factor 1 (TFF1) and *H. pylori* [94] or matrix metallopeptidase 9/neutrophil gelatinase-associated lipolalin (MMP-9/NGAL) complex [95] and a decrease in endothelial lipase (EL) in GC urine were promising diagnostic markers of GC. The high heterogeneity reported for plasma/serum proteomics also emerges when considering the results obtained with other fluid biological matrices (e.g., ascitic fluid, gastric juice, saliva, and urine) (Table 2).

## 3. Glycosylation of Circulating Proteins for GC Diagnosis

Another growing field of interest in biomarker discovery applied to GC diagnosis is protein glycosylation, a common post-translational modification occurring in over 50% of human proteins. Glycoproteomics focuses on the analysis of peptides with attached glycans (glycopeptides) and, via targeted approaches, is aimed at deciphering site-specific glycan distributions of extraordinarily complex glycoproteins. Many pieces of evidence have shown that glycosylation is closely connected with cancer development.

Protein glycosylation closely reflects the physiological state of the cell and can be affected by GC [130,131] (as recently reviewed in gastrointestinal tumors [132]). In the process of gastric mucosa malignant transformation, N-acetylglucosaminyltransferase-V glycosylates E-cadherin and integrin rapidly increase β1, 6-GlcNAc branched N-glycans [133,134]. It decreases cell–cell, and cell–extracellular matrix adhesive properties and promotes cancer cell invasion and metastasis.

Patients with precancerous gastric lesions present several circulating serum glycoproteins carrying abnormal O-glycans (e.g., plasminogen, vitronectin, and IGH protein), candidate targets for the non-invasive diagnosis of precursor GC lesions [135]. Along with disease progression, glycosylation affects proteins involved in complement activation, possibly due to the host’s response to the presence of the stomach tumor, and in acute phase response signaling, possibly due to increased signaling of the pro-inflammatory cytokine, IL-6 [130].

Three glycopeptides discriminating GC from C groups (AUC = 1.0, sensitivity and specificity = 100%) have been discovered by Lee et al. by creating an analytical platform with a targeted glycoproteomic approach (target protein-specific, glycosylation site-specific and structure-specific) to identify and quantify glycopeptides linked to serum haptoglobin (Hb), a major acute-phase highly sialylated glycoprotein composed of four N-glycosylation sites [136,137]. Aberrant Hb glycosylation in patients with GC was previously investigated in terms of N-glycan variation based on intact m/z signals: the AUC values of six combined glycan markers reached 0.8~0.93, and a diagnostic value of this multi-biomarker panel was evidenced for the first time [138]. Specific glycomic profiling of targeted serum Hb levels associated with GC was performed by Lee et al. [139]: Hb glycans highly branched and decorated with fucosylation and sialylation were found to be correlated with GC, and antennae fucosylation in tri- and tetra-antennary sialylated complex type N-glycan was the leading GC-associated glycan signature. The detection of abnormal serum haptoglobin glycosylation has gained increasing attention as a promising alternative approach to GC diagnosis/detection. Various assay diagnosis platforms (e.g., glycan, site-specific glycopeptide, and intact protein profiling) have been introduced, and an increase in specificity and sensitivity for clinical use still represents the main analytical challenge [140].

Altered glycosylation signatures associated with GC have also been reported for serum immunoglobulin G (IgG). Disease-specific IgG Fc N-glycosylation resulted in personalized biomarkers differentiating GC from benign gastric diseases. In particular, the G2FN/G1FN ratio discriminated female BGD patients from female GC patients in the age range of 20–79 years (sensitivity = 82.6%, specificity = 82.6%, and AUC = 0.872) [141]. A potential predictive power for the altered patterns of IgG glycosylation emerged in GC detection since they discriminated against patients affected by GC, duodenal ulcers, or non-atrophic gastritis [142].

Moreover, a decrease in IgG1 and sialylation and an increase in IgG4 mono-galactosylation were found in GC and esophageal and colorectal cancers, along with disease progression and inflammatory activities, with subclass-specific changes in all gastrointestinal cancers. The spatial and temporal diversity of IgG N-glycome among digestive cancers has been observed. IgG1-H5N5, IgG2-H4N3F1, and IgG4-H4N4F1 glycopeptides successfully discriminated all three cancer groups from the healthy controls [143].

Aberrant glycosylation patterns are known to occur in exosomes, including GC-related ones, in which an increase in Fucα1-6GlcNAc and Fucα1-3(Galβ1-4)GlcNAc has been recently detected using lectin microarrays [144]. Similar to what has been observed for prostate cancer, where the glycosylation patterns of exosomal prostate-specific antigen PSA correlated with disease state significantly better than the traditional PSA test [145], some glycosignatures of circulating exosomal proteins may serve as a basis for detecting GC.

Besides the enzymatic reaction of glycosylation, the other reaction of glucose and its metabolites with biological molecules, including proteins, is non-enzymatic glycation. Glycation may impair protein function/stability and induce the synthesis/activation of pathogenetic molecules—the intracellular protein high-mobility group box protein 1 and protein S100 that bind to the receptor for advanced glycation products (RAGE)—participating in many inflammatory and metabolic events, thus activating intracellular signaling mechanisms linked with cancer initiation [146]. The RAGE axis activation is known to contribute to GC development [147,148].

When glycation occurs with oxidation, the resulting combined process is often named glycoxidation. Following glycoxidation, proteins may denature, fragment, aggregate, and/or alter/lose their biological function, and several signaling pathways (e.g., Nf-kB) may be activated, thus initiating inflammatory processes or apoptosis. Recently, the products of protein glycoxidation (i.e., tryptophan, kynurenine, and Amadori products) were colorimetrically/fluorometrically assessed with nitrosative stress parameters in the plasma/serum of GC patients and succeeded in differentiating patients with GC from healthy controls with high statistical power (sensitivity, specificity, and area under curve, AUC), thus emerging as potential diagnostic biomarkers of GC [149].

In addition to blood, significant alterations in glycopatterns were also investigated in saliva using lectin microarrays, which allowed for the development of two diagnostic models discriminating against GC and atrophic gastritis based on 15 candidate lectins with high diagnostic power [150].

The abundance levels of 14 different N-glycans were recently used to distinguish GC tissues from adjacent ones using machine learning integrated with mass spectrometry-based N-glycomics [151]. Experimental glycomics data were already combined with proteomic data and clinical and pathological information using a machine learning methodology (KEM^®^, Knowledge Extraction and Management, Ariana Pharma, Cambridge, MA, USA) to characterize the subgroups of GC patients, and a high potentiality of this integrated large biomarker dataset emerged for non-invasive GC diagnosis and prognosis [152].

## 4. Serum Protein Marker Currently Used for Gastric Preneoplastic Evaluation

Serological markers currently used for gastric preneoplastic evaluations consist of specific biomarkers (i.e., gastrin-17 and pepsinogen PG) and non-specific ones (i.e., carbohydrate antigen 199, CA724, and carcinoembryonic antigen).

Variations in PG abundance and PGI/II ratios, particularly, low serum PG I concentration ≤ 70 ng/mL and PG I/II ratio ≤3, indicate stomach mucosa atrophy, a risk factor for gastric tumorigenesis because it can progress into in situ carcinoma via intestinal metaplasia and dysplasia, and patients at a high risk of GC can be thus identified. PGI typically demonstrated a higher decrease than PGII, thus leading to a lower PGI/II ratio [81]. Furthermore, in the case of *H. pylori* infection, an increase in PGII concentration may further decrease the PGI/PGII ratio [153]. Gastrin-17, a major form of gastrin, is mainly secreted by the gastric G cells and stimulates the growth of gastric mucosal endocrine cells (parietal and enterochromaffin-like). A low level of gastrin-17 was reported as a biomarker for atrophic gastritis in the gastric antrum [154].

Yu et al. [155] measured the serum levels of PGI, PGII, and gastrin-17 using ELISA in 68 patients with chronic atrophic gastritis and 86 healthy individuals. Their study demonstrated a lower statistical power of gastrin-17 than the PGI, PGII, and PGI/II ratio, suggesting a higher clinical value of PG in screening chronic atrophic gastritis than gastrin-17. Furthermore, patients with autoimmune atrophic gastritis showed a substantial increase in their G17 levels [153]. In a prospective single-center clinical study including 25 GC patients out of 116 enrolled patients, Trivanovic et al. found that PGI ≤ 70 and PGI/II ratio ≤ 3.0 cut-off values reach accuracy, sensitivity, specificity, positive predictive values, and negative predictive values for GC diagnosis, thus proposing pepsinogen tests for population screening aimed at avoiding unnecessary invasive endoscopic procedures [156]. These cut-off points for PGI and PGI/II ratio are widely accepted for identifying patients at risk of GC in regions with a high GC risk, such as Japan and Korea. However, there has been controversy in the literature regarding the validity of this test, particularly when the GC incidence rate is low and moderate. Moreover, several studies employing different analytical technologies have reported varying sensitivities and specificities, along with different cut-off values. More recently, in a cohort of patients suffering from early GC and intraepithelial neoplasia, Yanan et al. reported a decrease in PGI and p27 and an increase in G-17 levels via the aggravation of severity, thus proposing those serum markers for the diagnosis of early GC [157]. In a study with 275 GC and 275 healthy patients enrolled, the risk classification of GC was improved by adopting new PG criteria (PGII ≥ 10 ng/mL or PGI/II ≤ 5) with the addition of an *H. pylori* antibody test and reduced instances of GC cases being misclassified as low risk [158].

In a recent study [159], a screening strategy called the DSC test was introduced to identify individuals at risk of GC in geographical areas with a medium risk of GC incidence, such as Italy, where the age-standardized incidence rate (ASIR) is less than 14 per 100,000 [160]. To validate this test, two cohorts of individuals from Veneto and Friuli-Venezia Giulia, Italy, were enrolled: a retrospective cohort of 500 individuals and a prospective cohort of 163 individuals referred for an endoscopy. The DSC test’s classification utilized factors such as age, sex, serum pepsinogen I and II, gastrin 17, and anti-*H. pylori* IgG concentrations. Based on the test results, patients were categorized into low-, medium-, and high-risk groups for GC. Gastroscopies were performed by gastroenterologists, biopsies were taken from standardized mucosa sites, and a pathologist assessed the results for diagnosis. The DSC test demonstrated a good level of accuracy (74.66%) and high specificity (a true negative rate), surpassing the sensitivity of the more commonly used PGI ≤ 70 and PGI/II ratio ≤ 3.0 test. Importantly, the results obtained from applying the DSC test on a prospective, non-selected cohort were comparable to those achieved in a region with a high GC incidence (ASIR > 20 per 100,000) [161]. The DSC test was thus suggested to be valuable in identifying patients for opportunistic GC screening in medium-risk regions. In cases where individuals received a positive DSC classification, further evaluation via gastroscopy and more rigorous endoscopic surveillance could enhance the identification of individuals at a higher risk of early-stage GC, potentially allowing for more effective preventive measures, including minimally invasive treatments. To date, there has been one prospective study that combined only the PGI ≤ 70 and PGI/II ratio ≤ 3.0 test in a population with a medium GC risk, primarily focusing on monitoring patients with precancerous lesions [162]. This study found that high-grade dysplasia or neoplasia only developed in patients with extensive precancerous lesions and a low PGI/II ratio ≤ 3 and/or an OLGIM stage (III–IV) during follow-ups, which occurred approximately 57 months later. In De Re et al.’s study [159], after a median follow-up of 15.5 months, two out of 17 individuals experienced an elevation in DSC classification from the negative category to the neutral category, even though the histological diagnosis remained at moderate atrophy (OLGA stage 0–II).

The serological examination of GC may take advantage of the non-invasive multi-index-combined detection to enhance the selection of patients for upper gastrointestinal diagnostic endoscopy. However, it is crucial to recognize that GC predominantly occurs in older individuals, and aging is linked to a gradual decline in the integrity of gastric tissues, resulting in impaired function and changes in the PGI/PGII ratio. This age-related factor could potentially lower the precision of the pepsinogen test in individuals aged over 75 years. Consequently, it is essential to account for this factor when interpreting test outcomes. Therefore, in the future, incorporating additional biomarkers may be needed to improve the DSC test’s accuracy in older patients and minimize the likelihood of false positive results.

Nonetheless, the overall results indicated that the serological gastric function strategy, characterized by stringent prescription controls, proved to be effective in enhancing the appropriateness of patient selection for upper gastrointestinal endoscopy not only in regions with high GC incidence but also in medium-risk regions, as also further confirmed in a recent study [163], while the management of *H. pylori* was found to be useful in reducing GC development [164] and gastrin G17 in the diagnosis of autoimmune atrophic gastritis [165,166].

## 5. Circulating Exosomal Proteins

Over the last few decades, a new frontier in biomarker discovery for human diseases has been represented as exosomes and their interplay with cancer [167,168,169]. Exosomes represent a biological material sampled via minimally invasive liquid biopsy and provide useful information for disease diagnosis [168]. Exosomal protein application as biomarkers in GC diagnosis may have a lower cost and cause less pain than conventional diagnostic methodologies. Exosomes are extracellular nanoscale vesicles (30–150 nm) of endocytic origin that transport various biomolecules (i.e., proteins, glycans, lipids, metabolites, RNA, and DNA) [170]. The content of human exosomes is available in public online databases, such as ExoCarta (www.exocarta.org; 41,860 exosomal protein entries in the latest version, accessed on 22 November 2023) and Vesiclepedia (http://www.microvesicles.org; 566,911 extracellular vesicle protein entries in the latest version, accessed on 22 November 2023). Exosomes secreted via donor cells into extracellular spaces can be internalized in recipient cells, thus mediating cell–cell communication and compound exchange (i.e., soluble/insoluble signaling factors, proteins, nucleic acids, and lipids) and interfering with various physiological and pathological processes (angiogenesis, coagulation, proliferation, and senescence) [171]. Exosomes can be stably found in multiple biological fluids (e.g., blood, urine, and saliva), thus carrying functional information to distant sites. Exosomal protein cargo is extremely varied because exosomes are produced by almost all types of cells, and they reflect the identity of the originated cells. In general, the nature and abundance of exosome molecular cargo are closely influenced by intracellular changes occurring under different physiological and pathological conditions, including cancer [172]. Tumor-derived exosomes may transport tumor-associated bioactive molecules, such as mRNAs, microRNAs, and proteins, and, therefore, contribute to malignancy-related events (e.g., microenvironment reconstruction, angiogenesis, tumorigenesis, epithelial–mesenchymal transition, metastasis, and immune escape) [169]. Moreover, exosomes can transport messages between primary tumor cells and the microenvironment of distant recipient organs via bodily fluids, such as blood [173]. Therefore, exosomes isolated from cancer liquid biopsy are emerging as a revolutionary strategy for non-invasive cancer diagnoses [168].

The first evidence of a role played by tumor-derived exosomes in GC proliferation came from Qu et al. [174] via the activation of the PI3K/Akt and MAPK/ERK pathways. At present, GC cell-derived exosomes are known to be involved in various steps of GC development (e.g., tumorigenesis, metastasis, angiogenesis, immune evasion, and drug resistance) [175]. Although proteins are one of the major components of exosomes, knowledge of dynamic exosomal protein cargo is still in the early stages. Several tumorigenic exosomal proteins have been described in GC cells (i.e., LSD1, PD-L1, [176], EGFR [177], ApoE [178]). In this section, we discuss the proteins identified in exosomes isolated from the blood of GC patients. Compared to works focusing on circulating exosomal non-coding RNAs in GC [179,180], few studies have specifically evaluated plasma/serum exosomal proteins potentially implicated as diagnostic biomarkers for GC.

Exosomal proteins can have an important role in GC diagnosis [181]. Fu et al. [181] used the mass spectrometry protein profiles of exosomes extracted from the serum of GC patients and a cell culture supernatant and found a decreased level of tripartite motif-containing 3 (TRIM3), a member of the TRIM subfamily of the RING-type E3 ubiquitin ligases, in the serum of GC (n = 80) and healthy patients (n = 80). Lower contents were also observed in the tissue and validated using ELISA and WB. Although the observed GC-associated decrease in exosomal TRIM3 may contradict the role of TRIM3 overexpression in GC growth and metastatic spread, the authors concluded that TRIM3 may represent a diagnostic biomarker for GC.

Yoon et al. [182] found a much lower serum concentration of GKN1 in GC patients (n = 500) than in healthy individuals (n = 200), with a serum GKN1 diagnostic accuracy of 0.9675 at the optimum cut-off. Moreover, this cancer-related decrease in serum GKN1 was more evident in advanced GC patients than in early GC patients, with GKN1 diagnostic accuracies at the optimum cut-off (0.9675) of 0.8912 and 0.9589 for early GC and advanced GC, respectively. All of this evidence has demonstrated the specificity and candidate role of serum GKN1 as a biomarker of GC. Human GKN1 plays a pivotal role in maintaining mucosal homeostasis and regulating cell proliferation and differentiation. Yoon and colleagues demonstrated the exosomal nature of serum GKN1 internalized in gastric epithelium via an exosome-driven chlatrin-mediated transfer [183]. The exosomal form of GKN1 was found to suppress tumor growth in vivo and has thus been proposed as a therapeutic target of GC. A decreased gastric mucosa expression of GKN1 in patients with GC is known to promote gastric tumorigenesis [184]. In addition, serum exosomal GKN1 concentrations discriminated patients with early GC (n = 140) from healthy individuals (n = 200) (AUCs = 1.0000 and 0.9892, respectively), thus reinforcing the diagnostic value of serum GKN1 in GC [66].

Another key protein identified in the plasma exosomal cargo of patients with GC is the human leukocyte antigen G (HLA-G), an immune checkpoint molecule. Its high expression in cancer is associated with immune escape, metastatic spread, poor prognosis, and low overall survival. The first evidence of HLA-G in the cargo of exosomes enriched from the plasma of patients with gastrointestinal diseases, including GC, comes from Farjadian and colleagues [185], who observed significantly higher plasma sHLA-G levels in patients with gastrointestinal cancers (n = 82) compared with healthy controls (n = 45). In agreement with these data, Mejía-Guarnizo et al. [43] found HLA-G molecules in exosomal membranes and demonstrated the importance “to perform studies with a larger number of samples to explore the functional implications of HLA-G positive exosomes in the context of GC, and to determine the clinical significance and possible applications of these findings in the development of non-invasive diagnostics”. Higher HLA-G levels in GC patients (n = 81) than in patients with benign gastric disease (n = 53) and normal controls (n = 77) were also observed by Pan et al. [50], who also proposed detecting sHLA-G with other cancer markers (CA125 + CA19-9 + sHLA-G or CA125 + CA724 + sHLA-G) for the diagnosis of GC. These data highlight the importance of sHLA-G levels as a potential sentinel protein for GC diagnosis.

Coban et al. showed higher mean serum TGF-β1 levels in patients with GC (n = 32) and colon cancer (n = 36) than in a control group (n = 25) (*p* = 0.001) [186]. The TGF-β1 had higher sensitivity in patients with GC compared with those with colon cancer. Moreover, the TGF-β1 sensitivity was better than that for CEA in patients with GC.

Interestingly, in serum-derived exosomes from four GC patients infected with CagA-positive *H. pylori*, Shimoda et al. [187] detected the protein CagA, a major *H. pylori* virulence factor encoded using the cytotoxin-associated gene A. CagA-positive exosomes determined morphological modifications in gastric epithelial cells and GC cells, suggesting a link between functional CagA exosome delivery into cells and the development of extragastric disorders associated with CagA-positive *H. pylori* infection.

Recently, serum-derived exosomal HER2 was found to be a highly specific sentinel molecule to assess tissue HER2 status, with a stable diagnostic effect in patients with advanced GC (n = 238, of which 114 were HER2-positive), and thus screened patients that could potentially benefit from anti-HER2 therapy [188].

## 6. Conclusions

Over the last few years, recent advancements in the molecular characterization of the inter- and intra-tumor heterogeneity of GC in many individuals have forced researchers to gain better insight into the hallmarks associated with the early phases of GC at different levels, including proteins. Diagnostic biomarkers should be specific protein markers, individual or combined, that are able to improve early GC diagnosis, reflecting both interpatient and tumor heterogeneity to be applied to a personalized medicine scenario. In cohorts of patients differing in both size and clinical features, with different biological fluids, and using different instrumental/analytical approaches (mainly ELISA and MS), an increasing number of circulating proteins have been analyzed (Appendix A). Most targeted works investigated different protein analytes and most untargeted works identified different putative diagnostic biomarkers. Heterogeneity in statistical methods may also account for differences across studies. Univariate analysis is often used to compare protein levels in GC patients and non-cancer patients. Generalizability should be improved by the adoption of multivariable models, which take into account differences across study populations in socio-demographic, lifestyle, and clinical characteristics and that could impact protein abundance. However, sample sizes are often small, limiting the adoption of multivariable models.

Proteomics biomarker discovery applied to GC research has succeeded in the identification of potential predictive diagnostic biomarkers (i.e., HLA-G, IL-6, PD-1), evidenced aberrant GC disease-related glycosignatures (i.e., N-glycosylation), and found in circulating exosomal cargo a new important source of diagnostic markers (i.e., TRIM3, GKN1, and HLA-G). Several serum protein panels successfully used for gastric preneoplastic evaluation focused on PGI, PGII, PGI/II ratio, and gastrin-17 levels and succeeded in finding optimal cut-off values with high accuracy, sensitivity, specificity, and predictive values. Overall, these markers would guide clinicians and physicians to better manage patients and to a circuit leading to the characterization of cancer molecular profiles (i.e., through analyses of tissue/liquid biopsy) and the selection of a patient-tailored therapy.

Some GC diagnostic biomarkers discovered by proteomics approaches are promising, but few have been extensively validated in large cohorts of patients. Therefore, besides the increasing interest in finding new biomarkers, several efforts should be addressed to the validation of those recently identified. Proper experimental designs, standardized procedures and quality controls for sample collection and analyses, and correct validation phases are necessary to test the sensitivity, specificity, and reproducibility of clinically relevant biomarkers. Some issues about biomarker development have been recently highlighted by excellent reviews [97].

Gastric cancer disease is very heterogeneous with different clinical outcomes, so GC biomarker discovery and validation for a real clinical application are particularly arduous. Globally, the identification of clinically useful markers is very hard because of the high complexity of biological samples, especially plasma for its high dynamic range, inter- and intra-patient variability, and lack of analytically sensitive techniques for both discovery and validation. Although the circulating markers may be of great utility in developing non-invasive tools for an early detection of GC, they seem not to be sufficient for early accurate detection, especially when an individual protein biomarker is used.

Some biomarkers have shown increased diagnostic accuracy when combined into protein biomarker panels and with clinical data using dedicated algorithms. At present, despite the availability of various proteomic techniques measuring biomarker panels, the integration of proteomics into clinical practice has been limited. In particular, many challenges need to be addressed for the discovery of the most promising protein biomarkers and their application to clinical practice (recently commented by [189]). Several developments in MS-based approaches, ranging from sample preparation to bioinformatics tools, were successful in bringing proteomics closer to clinical application (reviewed by [42]), as demonstrated by the presence of some FDA-approved cancer biomarkers based on targeted proteomics (i.e., serum OVA1, in vitro diagnostic multivariate index assay by SELDI-TOF-MS in ovarian cancer [190]). Recently, the combination of proteomics data with those obtained with other omics methods (genomics, epigenomics, transcriptomics, and metabolomics) in the so-called “multi-omics approach” with advancements in machine learning algorithms has recently shown the potential for interesting applications in cancer research [191]. A machine learning approach has been recently adopted to distinguish GC from control tissues with high accuracy after integration with mass spectrometry-based N-glycomic data [151]. A great challenge for biomarker discovery applied to GC prediction might thus come by integrating protein profiling data (qualitative and quantitative) with different approaches to be translated into tools that are accessible for routine clinical applications. These advanced tests should be accessible and affordable to reach the greatest healthcare benefit.

## Figures and Tables

**Figure 1 ijms-24-16931-f001:**
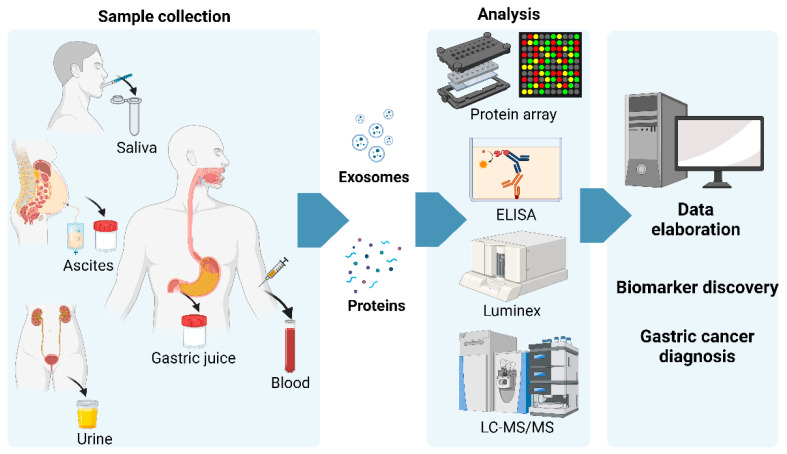
Schematic illustration of analytical workflow for biomarker discovery. Saliva, ascites, urine, gastric juice, urine, and blood are collected; proteins/exosomes are enriched and analyzed; data are elaborated; and putative biomarkers are discovered. Their abundances are usually compared with clinical information to achieve early detection. Created with BioRender.com, accessed on 31 October 2023.

**Figure 2 ijms-24-16931-f002:**
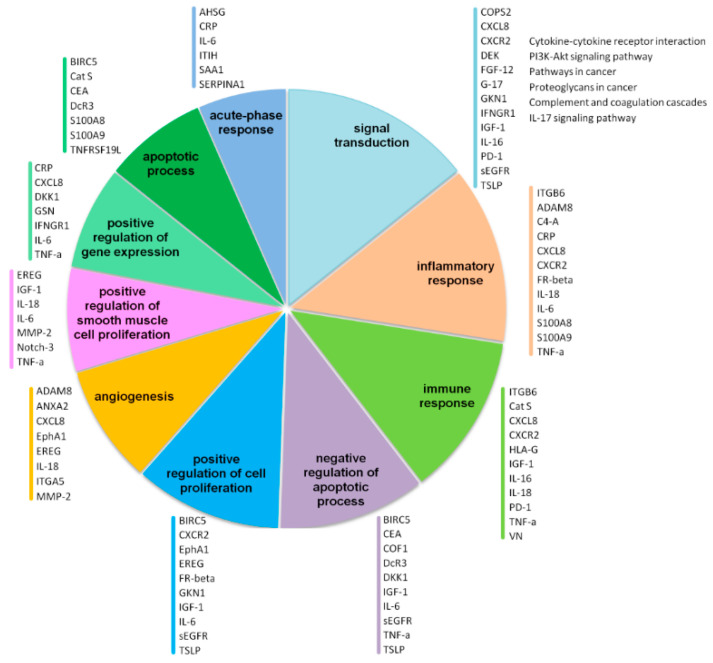
Diagram showing the top 10 most significant Gene Ontology (GO) biological processes of plasma/serum proteins found to be associated with gastric cancer diagnosis in the last 10 years (Table 1; *p* < 0.05; FDR < 0.05). The diagram results from the interrogation of proteins listed in Table 1 with DAVID 6.8 (https://doi.org/10.1038/nprot.2008.211, accessed on 11 September 2023). For each GO biological process, the list of involved proteins is reported (UniProtKB entry protein; https://www.uniprot.org/, accessed on 11 September 2023). KEGG pathways are listed next to the signal transduction bia Alikhani ological process.

**Table 1 ijms-24-16931-t001:** List of blood-based protein markers for GC diagnosis reported over the past 10 years.

Biomarker(s) ^(a)^	GO Biological Process ^(b)^	Proteomic Technology ^(c)^	Patients		Major Findings	Ref.
			Tumor Characteristics ^(d)^	Group (*nr*) ^(e)^		
** *Plasma* **						
sHLA-G [P17693] (plasma and exosomal)	Immune response	• ELISA	• Histology: 58.4% intestinal, 26.0% single ring cell adenocarcinoma, 15.6% other • clinical stages: 24.9% I–II, 56.6 III–IV • 39.9% poorly differentiated	GC (173), benign gastric disease (307; 86.3% chronic gastritis)	A higher sHLA-G concentration was found in GC vs. benign pathologies in GC-affected women vs. men, but no significant differences were found among the GC stages. sHLA-G was proposed as a potential diagnostic marker, although not as an adequate marker for staging GC. HLA-G was found in exosome membranes.	[43]
18 proteins	Miscellaneous	• LC-MS/MS combined with TMT labeling	• Early-GC • adenocarcinoma (87%) and high-grade intraepithelial neoplasia (13%) • adenocarcinoma mainly well or moderately differentiated with invasive depth mainly limited to the mucosa	early-GC (15) and C (15)	From a total of 2040 proteins identified, 11 proteins were differentially abundant between early-GC patients and C (7 increased and 4 decreased). These proteins distinguished early-GC from healthy C (sensitivity = 66.7%, and specificity = 86.7%).	[44]
PD-1 [Q15116] PD-L1 [Q9NZQ7] macrophage and B-cell markers	Adaptive immune response Adaptive immune response	• ELISA • IHC	• Histology: 82.5% adenocarcinoma, 16% signet ring cell carcinoma, 1.5% undifferentiated cancer • clinical stages: 40% I–II, 60% III–IV • depth of invasion: 21% T1–T2, 79% T3–T4 • metastasis: 86% M0, 14% M1	GC (63)	The plasma content of the sPD-1 receptor was significantly lower in GC vs. C; it inversely correlates with plasma sPD-L1 content and directly correlates with the tissue PD-L1 expression in stromal cells. Levels of sPD-L1 in GC vs. C were similar.	[45]
TrxR [Q86VQ6]	Cell differentiation	• Ultraviolet spectrophotometry • ECLA	• Adenocarcinoma	GC (896), benign gastric disease (322; e.g., stomach ulcer, stomach polyps, and gastritis) and C (228)	TrxR activity in GC [8.4 U/mL] was significantly higher than that in benign disease [6.1 U/mL] or C [3.7 U/mL]. ROC analysis of TrxR [AUC = 0.945; sensitivity = 95.6%; specificity = 76.3%] showed a better capacity of GC diagnosis than that of routine tumor markers (AFP, CA50, CA72-4, CA19-9, CA242, CEA).	[46]
sEGFR [P00533] TSLP [Q969D9]	Cell morphogenesis/adhesion Positive regulation of chemokine production	• 5 Luminex bead-based multiplex assay panels	• Clinical stages: 52.5% early/localized, 34.1% advanced, 13.4% unknown	GC (446) and individuals (774) as random subcohort.	Levels of sEGFR (*P_t_*_rends_ = 0.017) and TSLP (*P_t_*_rends_ = 0.034) were associated with GC risk. However, none of the *P_t_*_rends_ remained statistically significant after FDR correction.	[47]
DEK [P35659]	Chromatin remodeling	• WB • ELISA	• Histology: 72% diffuse, 28% intestinal • depth of invasion: 35% T1–T2, 65% T3–T4 • GC undoing gastrectomy • 39% localized, 61% infiltrative • 67% lymph node metastasis, 18% distant metastasis.	GC (92) and C (120)	Data from ROC curve analysis highlighted a better diagnostic accuracy (AUC = 0.797) and sensitivity (70.4) than CEA, CA 19-9, and CRP.	[48]
ApoC1 [P02654] GSN [P06396] SHBG [P04278] C4-A [P0C0L4]	Cholesterol efflux Actin filament capping Androgen binding Complement activation	• Label-free quantitative LC-MS/MS • bioinformatics • WB • ELISA	• Adenocarcinoma • histology: *discovery cohort*→25% diffuse, 71% intestinal, 4% mixed; *verification cohort*→29% diffuse, 71% intestinal; *validation cohort*→52% diffuse, 40% intestinal, 8% mixed • depth of invasion: *discovery and verification cohorts*→50% T1–T2, 50% T3–T4; *validation cohort*→34% T1–T2, 66% T3–T4	*Discovery cohort*: GC (24) and C (9); verification *cohort*: GC (24) and C (9); *validation cohort*: GC (50) and C (68)	Four proteins (apolipoprotein C-1, gelsolin, SHBG, and complement component C4-A) increased in content in GC (*p* < 0.05). WB and ELISA confirmed higher SHBG levels in GC. Plasma SHBG levels were proposed as a potential early diagnostic biomarker for GC.	[49]
sHLA-G [P17693]	Immune response	• ELISA	• Histology: 37% diffuse, 49.3% intestinal, 13.7 mixed • clinical stages: 38% I–II, 62% III–IV • depth of invasion: 25.9% T1–T2, 74.1% T3–T4 • distant metastasis: 95% M0, 5% M1	81 GC, benign gastric disease (53, e.g., ulcer, gastritis, polypus) and C (77)	Plasma sHLA-G concentration was significantly higher in GC compared with both benign gastric disease and C. sHLA-G was proposed as a GC diagnostic marker, especially when combined with other GC markers (CA125, CA19-9, and CA72-4).	[50]
** *Serum* **						
TNFα [P01375] IL-8 [P10145]	Acute inflammatory response Angiogenesis	• ELISA	• Histology: 10% diffuse, 48% intestinal, 22% signet ring cell, 12% mixed, 8% other types • various clinical stages	GC (82), CG (94), and C (53)	Moderate levels of TNF-α were detected in the C group (19.9 ± 19.5 pg/mL), which were significantly higher in CG patients (35.7 ± 28.0 pg/mL) but drastically decreased in GC (1.8 ± 5.9 pg/mL). TNF-α was proposed to behave as an inflammatory marker. IL-8 concentrations did not vary among patients.	[51]
ITGB6 [P18564] GPX3 [P22352] CRP [P02741] S100A9 [P06702] SERPINA4 [P29622]	Cell adhesion/morphogenesis hydrogen peroxide catabolic process Acute-phase response Apoptotic process Negative regulation of endopeptidase activity	• LC-MS/MS	• Histology: diffuse • WHO classification: 61% tubular, 39% poorly cohesive	219 *H. pylori* positive and negative patients diagnosed with GC, gastritis, and ulcers	Two GC serum marker panels, 29preGC-P (with ITGB6 and GPX3) and 10GC-P (with CRP, S100A9, and SERPINA4), were proposed for the diagnosis of early stage and advanced GC independently on *H. pylori* status, respectively.	[52]
G-17 [P01350] PGI [P0DJD8] PGII [P20142] PGR [P06401]	Response to food Digestion Digestion Cell–cell signaling	• GastroPanel ELISA kit	• 75.0% adenocarcinoma, 11.1% mucinous carcinoma, 8.3% poorly cohesive, 2.8% tubular carcinoma, 2.8% papillary carcinoma	GC (36), AG (40), and C (40)	PGI levels significantly decreased in GC and AG compared to C groups (*p* < 0.05). No significant differences in PGII and G-17 levels between study groups. For GC, the optimal cut-off values of PGI and PGR (PGI to PGII ratio) were ≤35.25 ng/mL (sensitivity = 47.2%; specificity = 86.8%) and ≤5.27 ng/mL (sensitivity = 75%; specificity = 60.5%), respectively. The PGR was significantly lower in GC vs. C (*p* < 0.01). The combinations of PGI and PGR with risk factors were proposed to improve diagnostic accuracy (AUC for AG 74.8, 95% CI 64.0–85.7, *p* < 0.001; AUC for GC 75.5, 95% CI 64.2–86.8, *p* < 0.001).	[53]
IGF-1 [P08069]	Insulin receptor signaling pathway	• ELISA • IHC • real-time PCR assay	• Clinical stages: 39.3% I–II, 60.7% III–IV • invasion depth: 30% T1–T2, 38.4% T3–T4, any T 31.6% • distance metastasis: 68.4% M0, 31.6% M1 • *H. pylori* status: 69.6 pos, 30.4% neg	GC (60) and C (30)	Early GC stages showed a significantly low IHC score for IGF-1R and phosphorylated AKT, mTOR, and ERK proteins compared to the advanced stages. IGF-1 serum levels and the expression of candidate genes increased in advanced vs. early GC and positive vs. negative *H. pylori* status (*p* < 0.05).	[54]
IL-6 [P05231] PGI [P0DJD8] PGII [P20142] PGR [P06401] TNF-α [P01375]	Acute-phase response Digestion Digestion Cell–cell signaling Acute inflammatory response	• ELISA • IMMU-NITE1000	Not detailed	*Observation group*: GC (50) with *H. pylori*; *comparison group*: GC (50) without *H. pylori*.	In the “observation” group, PGI and PGII were lower, while TNF-α, IL-18, and IL-6 were significantly higher than those in the “comparison” group (*p* < 0.05).	[55]
CEA [P06731] SAA [P0DJI8] IL-6 [P05231]	Apoptotic process Acute-phase response Acute-phase response	• Commercial kits	Not detailed	GC (122), gastric benign disease (37), and C (30)	SAA and IL-6 levels were higher in GC vs. C. The ROC curve for the combined detection of SAA, IL-6, and CEA showed AUC = 0.948, sensitivity = 91.0%, and specificity = 89.2%.	[56]
SIRT6 [Q8N6T7]	DNA repair-dependent chromatin remodeling	• ELISA	• Histology: 36.3% diffuse, 63.7% intestinal • clinical stages: 38.5% I–II, 60.7% III–IV • invasion depth: 25.9% T1–T2, 74.1% T3–T4 • distance metastasis: 85.9% M0, 14.1% M1	GC (135), AG (68), and C (60)	Serum SIRT6 levels were lower in GC vs. AG and C. They were positively associated with tumor stage and metastasis and proposed as a diagnostic and predictive biomarker for GC.	[57]
AFP [P02771] CA125 [Q8WXI7] CEA [P06731] CA153 [n.a.] CA199 [n.a.] CA242 [n.a.]	Progesterone metabolism Cell adhesion Apoptotic process	• Multi-tumor marker detection kit based on protein chips	Not detailed	GC (268) and C (209)	Serum GC was associated with age, gender, and positive levels of AFP, CEA, CA125, CA199, and CA242. The positive levels of AFP and CA125 were related to distant GC metastasis.	[58]
SNCG [O76070]	Regulation of neurotransmitter secretion	• ELISA	• Clinical stages: 45% I–II, 55% III-IV • invasion depth: 34% T1–T2, 66% T3–T4 • distance metastasis: 83% M0, 17% M1 • *H. pylori* status: 41% positive, 59% negative	GC (87), gastric precancerous lesions (38), and C (44)	Detection of SNCG in serum and gastric juice was a good method for the early diagnosis of GC (cut-off value = 7.716 ng/mL; AUC = 0.924; sensitivity = 95.40%; specificity = 86.36%; *p* < 0.0001). Serum SNCG was related to TNM stage, lymph node metastasis, and tumor size.	[59]
DKK1 [O94907] TK1 [P04183] CA724 [n.a.]	Cell morphogenesis Mitotic DNA replication	• ELISA • ECLIA • ECLA	• GC without any type of treatment • clinical stages: 77.8% I–II, 22.2% III–IV	GC (63) and gastric benign disease (considered as C; 54)	The three serological indexes were higher in GC vs. C (*p* < 0.001). The ROC analysis for their combined detection showed an AUC = 0.923, with sensitivity and specificity higher than those of separate detection.	[60]
PD-1 [Q15116] PD-L1 [Q9NZQ7]	Adaptive immune response Adaptive immune response	• ELISA	• GC without any type of treatment • clinical stages: 66.6% I–II, 33.4% III–IV • patients undergoing gastrectomy and lymph node dissection	GC (30) and C (30)	Preoperative sPD-1 and sPD-L1 were lower in GC vs. C. The ROC analysis showed an AUC equal to 0.675 and 0.885 for sPD-1 and sPD-L1, respectively.	[61]
CXCL8 [P10145] CXCR2 [P25025] CEA [P06731] CA19.9 [n.a.]	Angiogenesis Immune response Apoptotic process	• ELISA • CMIA • turbidimetric assay	• Histology: 53% intestinal, 47% diffuse • clinical stages: 28% I–II, 69% III–IV, 3% undefined • invasion depth: 15.6% T1–T2, 84.4% T3–T4 • distant metastasis: 72% M0, 28% M1	GC (64) and C (34)	Higher levels of CXCL8 and CXCR2 in GC vs. C. Serum CXCL8 was proposed as a promising biomarker for GC diagnosis, especially in combination with CA19-9 (sensitivity = 89%; specificity = 53%).	[62]
ITIH4 [Q14624]	Acute-phase response	• LC-MS/MS • WB • IHC	• Clinical stages: 53% I–II, 47% III–IV	Early-GC (38), advanced GC (70), LGN (28, precancerous group), CSG (37), OST (49, patients with other system malignant tumors), and C (178)	ITIH4 abundance in early GC (specificity = 94.44%) was significantly higher than those in the C and other GC groups. ITIH4 was proposed as a diagnostic marker for early-GC.	[63]
SOX3 [P41225]	Cell differentiation	• LC-MS/MS combined with TMT/iTRAQ labeling	• Locally advanced GC (55 cases) and early GC (5 cases) • invasion depth: 31.7% T1–T2, 68.3% T3–T4 • clinical stages: 33.3% I–II, 66.7% III–IV • degree of differentiation: 26.7% poorly, 31.7% moderately, 41.6% well • distant metastasis: 90% M0, 10% M1	GC (60) and C (60)	Among proteins significantly differentially abundant, SOX3 was found to be higher in GC vs. C sera.	[64]
19 proteins	Miscellaneous	• PEA (over 300 proteins tested)	• Clinical stages: 28% I–II, 71% III–IV, 1% undefined	GC (100) and C (50)	In total, 19 serum proteins distinguished GC from C, with a diagnostic sensitivity of 93%, specificity of 100%, and AUC of 0.99 (95% CI: 0.98–1). This protein signature increased diagnostic capacity, particularly in patients at TNM I-II stages (sensitivity = 89%; specificity = 100%; AUC = 0.99) and with high microsatellite instability (MSI) (91%, 98%, and 0.99) compared to individual proteins.	[65]
GKN1 [Q9NS71]	Digestion	• ELISA	Not detailed	Early GC (140), advanced GC (360), other cancers (768), and C (200)	Serum GKN1 levels in GC (median: 3.48 ng/μL) were lower than in C (median: 6.34 ng/μL). GKN1 levels were significantly higher in early GC (median: 4.31 ng/μL) than in advanced GC (median: 3.11 ng/μL). The ROC curve analysis distinguished early from advanced GC (AUC = 0.870). Serum GKN1 appeared as a promising and highly specific diagnostic biomarker for both early and advanced GC.	[66]
EphA1 [P21709] EREG [O14944] FGF-12 [P61328] FR-β [P14207] Galectin-8 [O00214] GHR [P10912] IFNGR1 [P15260] Integrin α 5 [P08648] Notch-3 [Q9UM47] SLAMF8 [Q9P0V8] TNFRSF19L [Q969Z4]	Angiogenesis Angiogenesis Cell–cell signaling Cell adhesion Lymphatic endothelial cell migration Cytokine-mediated signaling pathway Positive regulation of gene expression Angiogenesis Notch signaling pathway Signaling receptor activity Apoptotic process	• Human cytokine antibody • ELISA	• Clinical stages: IA • undifferentiated • tumor location: mucosa • no metastasis	Antibody array assay: GC (15) and C (10); ELISA: GC (20) and C (20)	In total, 11 serum cytokines increased in GC (*p* < 0.05) and were proposed as novel biomarkers for the early diagnosis of GC.	[67]
BIRC5 [A0A7L8XZM3]	Regulation of apoptotic process	• ELISA	Not detailed	10.4% together with prostate cancer and glioblastoma from 67 patients with diagnosed cancer	Serum survivin level was high at GC diagnosis (*p* < 0.05). The optimal cut-off value of serum survivin was determined at >120.8 pg/mL.	[68]
CD59 [P13987] COF1 [P23528] S100A8 [P05109] ITIH4 [Q14624]	Blood coagulation Actin cytoskeleton organization Acute inflammatory response	• TMT labeling • LC-MS/MS and bioinformatics • WB	•Adenocarcinoma • stages: I/II	GC (10) and C (10)	A total of 105 proteins differed (*p* < 0.05) between GC and C, 69 being glycoproteins. The decrease in COF1 and the increase in ITIH4, S100A8, and CD59 could be useful in GC diagnosis.	[69]
IL-8 [P10145 ] TNF-α [P01375] CEA [P06731] IL-6 [P05231] CA72-4 [n.a.]	Angiogenesis Acute inflammatory response Apoptotic process Acute phase response	• Luminex 200 bead-based assay for IL-6, IL-8, and TNF-α determination • electrochemiluminescence	• Clinical stages: *discovery cohort* →36% early stage and 64% advanced stage; *validation cohort*→33% early stage and 67% advanced stage	*Discovery cohort*: GC (176) and C (204); *validation cohort*: GC (58) and C (66)	Serum IL-6 had the best diagnostic value in discriminating GC (AUC of joint analysis = 0.95). The multiparameter model with CEA, CA72-4, IL-6, IL-8. and TNF-α discriminated GC, early GC, and advanced GC from the healthy C one (sensitivity = 89.66%, 84.21%, and 92.31%; specificity = 92.42%, 90.91%, and 90.91%).	[70]
TK1 [P04183] CEA [P06731] CA19.9 [P15391] CA72-4 [n.a.]	DNA synthesis Apoptotic process Antigen receptor-mediated signaling pathway	• Cell Cycle Assay Kit • ECLIA assay kits	Not detailed	GC (169) and C (75)	TK1 was a good independent marker for GC. Its combination with CA19.9, CA72-4, and CEA performed better. The combined detection of the 4 markers was proposed to be useful for GC diagnosis.	[71]
AHSG [P02765] APOA-I [P02647] FGA [P02671]	Acute-phase response Blood vessel endothelial cell migration Blood coagulation	• MB-IMAC-Cu, MALDI-TOF-MS, and peptide pattern analysis • Nano Acquity UPLC MS/MS • ELISA	• Clinical stages: *discovery cohort*→50% I/II and 44% III/IV; *validation cohort*→38% I/II and 62% III/IV	*Discovery cohort*: GC (32) and C (30); *validation cohort*: GC (42) and C (28)	Among the 12 differential peptide peaks (*p* < 0.0001), the serum levels of FGA increased in GC vs. C (AUC = 0.98, *p* < 0.05), similar to AHSG and APOA-I (AUC = 0.92 and 0.83; *p* < 0.05), and these 3 proteins were proposed as valuable biomarkers for GC.	[72]
Leptin	Energy homeostasis, neuroendocrine function, and metabolism [Kelesidis 2010]	• ELISA	• Clinical stages: 16% I/II and 70% III/IV • invasion depth: 22% T1–T3 and 35% T4 • metastasis: 51% M0	GC (63) and C (30)	Leptin concentrations were lower in GC vs. C (*p* = 0.009). A diagnostic role was proposed for serum leptin levels.	[73]
CA19.9 [P15391] CEA [P06731] AFP [P02771] CA125 [Q8WXI7]	Antigen receptor-mediated signaling pathway Apoptotic process Progesterone metabolism Cell adhesion	Not reported	• Invasion depth: 43% T1a and 57% T1b	Early GC (587)	The positive rates of CEA, CA19.9, AFP, and CA125 (4.3%, 4.8%, 1.5%, and 1.9%, respectively) were low for early GC. The combination of CEA, CA125, and CA19–9 has been reported to lead to higher sensitivity than CEA alone.	[74]
DcR3 [O95407]	Apoptotic process	• ELISA	• Metastasis: 30% M0 and 70% M1	GC (10) and C (25)	DcR3 levels increased in GC patients (2.04 ± 1.01, *p* = 0.0061). ROC analysis showed high specificity (90%), sensitivity (85.7%), and AUC (82.3%; threshold = 243.7 pg/mL) to distinguish GC. DcR3 was proposed as a biomarker for GC diagnosis.	[75]
IL-2R [P01589] VEGF [P15692] TGF-β1 [P01137]	Activated T cell proliferation Angiogenesis ATP biosynthetic process/cell migration	• ELISA	• Clinical stages: 5.8% II, 94.2% III-IV	GC (35) and C (32)	Serum levels of sIL-2R, VEGF, and TGF-β1 were higher in GC vs. C. Serum sIL-2R levels were also positively associated with VEGF and TGF-β1 levels.	[76]
IL-16 [Q14005]	Cytokine-mediated signaling pathway	• ELISA • WB	Not detailed	GC (98) and C (98)	IL-16 levels in GC vs. C were higher (2.59-fold; *p* < 0.05). It differentiated GC from C (AUC = 0.882, sensitivity = 79.6%, and specificity = 78.6%). IL-16 was proposed as a novel diagnostic marker for GC.	[77]
CLU-1 [P10909] SRMS [Q9H3Y6] THB1 [P10828] VN [P04004]	Cell morphogenesis Cell differentiation Cell differentiation Cell adhesion	• Label-free quantitative LC-MS/MS • MS-based MRM • WB	• *Discovery cohort*: 50% early and 50% advanced; *validation cohort*: 52% early and 48% advanced	*Discovery cohort*: GC (6) and C (3); *validation cohort*: GC (60) and C (29)	In early GC, 119 and 176 proteins were up- and downregulated, respectively. Four proteins (VN, CLU-1, THB1, SRMS) changed in GC vs. C and discriminated GC with sufficient specificity and selectivity.	[78]
Cat S [P25774]	Antigen processing and presentation	• ELISA • WB • immunohistochemistry	• Clinical stages: 44.5% I–II, 55.5% III–IV	GC (119) and C (99)	Serum Cat S levels increased in GC (AUC = 0.803, sensitivity = 60.7%, and specificity = 90.0%).	[79]
COPS2 [P61201] CTSF [Q9UBX1] NT5E [P21589] TERF1 [P54274]	Negative regulation of transcription by RNA polymerase II Antigen processing and presentation of exogenous peptide antigen via MHC class II Adenosine biosynthetic process Cell division	• Proteome microarray • ELISA	• Invasion depth: *discovery phase I*→54% T1–T2, 46% T3–T4; *training/testing phase*→52% T1–T2, 48% T3–T4 • metastasis: 96% M0 and 4% M1	*Discovery phases I and II*→GC (37–300) and C (50–300); *training set*→GC (108) and C (108), *testing set*→GC (192) and C (192); *validation phases I and II*→ GC (100–200) and C (100–200)	A final panel of 4 biomarkers (COPS2, CTSF, NT5E, and TERF1) provided high diagnostic power (sensitivity = 95% and specificity = 92%) to differentiate GC from C, and it was proposed as a non-invasive diagnostic index for GC.	[80]
ADAM8 [P78325] VEGF [P15692] PGI [P0DJD8] PGII [P20142] IgG to *H. pylori*	Angiogenesis Angiogenesis Digestion Digestion	• Multiplex assay	• Newly diagnosed primary adenocarcinoma • invasion depth: 36% T1–T2, 64% T3–T4 • clinical stages: 23.1% I, 32.6% II, and 44.2% III; 17% early stage and 83% advanced stage	*Training set*: GC (228) and C (190); *validation set*: GC (48) and C (47)	The selected panel of markers differentiated between the majority of GC and C with high accuracy (RF 79.0%, SVM 83.8%, logistic regression 76.2%) in the training set as well as in the validation one (RF 82.5%, SVM 86.1%, logistic regression 78.7%).	[81]
FGA carboxyl-terminal fraction [P02671]	Coagulation	• SELDI ProteinChip analysis • LC-MS/MS • immunodepletion • chemiluminescence	• Invasion depth: *training set*→30% T1–T2, 70% T3–T4; *validation set*→20% T2, 80% T3–T4	*Training set*: GC (30) and C (30); *validation set*: GC (10) and C (10)	Peak 5910 showed good performance in distinguishing GC from C with high sensitivity and specificity (AUC = 0.89; *training set*→sensitivity = 86.3% and specificity = 91.3%; *validation set*→sensitivity = 100% and specificity = 93.3%).	[82]
ITIH4 [Q14624] SAA1 [P0DJI8]	Acute-phase response Acute-phase response	• iTRAQ labeling • SCX fractionation • LC-MS/MS • LC-MRM	• Histology: *discovery cohort*→100% at III; *validation cohort*→60% at III, 30% at II, 10% signet ring cell	*Discovery cohort*: GC (10) and C (10); *validation cohort*: GC (10) and C (10)	In GC, a total of 59 proteins were differentially abundant, 48 being up- (iTRAQ ratios of ≥2) and 11 being downregulated (iTRAQ ratios of ≤0.5). Validation analyses confirmed the increased levels of ITIH4 and SAA1 (*p* < 0.05) in GC.	[83]
IL-18 [Q14116]	Angiogenesis	• ELISA	• Clinical stages: 16% I-II, 70% III, and 14% undetermined • invasion depth: 22% T1–T3, 35% T4, and 43% unknown	GC (63) and C (30)	The baseline IL-18 levels of GC were higher than those of C (*p* < 0.001), indicating that IL-18 was a good serological diagnostic GC marker. No correlation was observed between IL-18 concentrations and clinical characteristics (*p* > 0.05).	[84]
ANXA2 [P07355]	Angiogenesis	• ELISA	• Clinical stages: 16% I-II, 70% III, and 14% undetermined • invasion depth: 22% T1–T3, 35% T4, and 43% unknown	GC (63) and C (30)	The baseline ANXA2 levels of GC were higher than those of the C group (*p* < 0.001). The known clinical variables were not correlated with ANXA2 concentrations (*p* > 0.05).	[85]
ANGPTL2 [Q9UKU9]	Cell–cell signaling	• IHC • ELISA	• Clinical stages: *screening phase*→50% I and 50% IV; *validation phase*→51% I, 16% II, 18% III, and 15% IV	*Screening phase*: GC (16) and C (23); *validation phase*: GC (194) and C (45)	Serum ANGPTL2 in GC was higher than in C (*p* < 0.05) and distinguished GC patients from C patients (AUC = 0.865). The validation step confirmed higher ANGPTL2 levels in GC vs. C (*p* < 0.0001).	[86]
SERPINA1 [P01009] ENOSF1 [Q7L5Y1]	Acute-phase response Amino acid/carbohydrate catabolic process	• WCX fractionation and MALDI-TOF MS • LC-MS/MS • ELISA	• Invasion depth: *discovery cohort*→9% I, 20% II, 46% III, and 25% IV (characteristics of patients belonging to the validation cohort not detailed)	*Discovery cohort*: GC (70); *validation cohort*: GC (36) and C (36)	Peptides with m/z values of 1546.02 and 5335.08 showed a higher concentration in the spectra of GC vs. controls (*p* < 0.001) and were identified as belonging to SERPINA1 and ENOSF1. Only ENOSF1 concentration was higher in GC (1.55-fold, *p* < 0.001) and it was proposed as a biomarker for GC diagnosis.	[87]

Abbreviations: ADAM8, adisintegrin and metalloproteinase domain-containing protein 8; AFP, α fetoprotein; AG, atrophic gastritis; AHSG, α-2-HS-glycoprotein; AKT, protein kinase B; ANGPTL2, angiopoietin-like protein 2; ANXA2, annexin A2; APOA-I, apolipoprotein A1; ApoC1, apolipoprotein C-1; AUC, area under curve; BIRC5, baculoviral IAP repeat containing 5 isoform 5 transcript variant 6 (surviving); C, healthy controls; C4-A, complement component; CA19.9, carbohydrate antigen 19.9; CA72-4, carbohydrate antigen 72-4; CA125, cancer antigen 125; CA153, cancer antigen 153; CA199, cancer antigen 199; CA242, cancer antigen 242; CA724, cancer antigen 724; Cat S, cathepsin S; CD-59, CD-59 glycoprotein; CEA, carcinoembryonic antigen; CLIA, chemiluminescence immunosorbent assay; CLU-1, clusterin isoform 1; CMIA, chemiluminescent microparticle immunoassay; COF1, cofilin-1; COP9, constitutive photomorphogenic homolog subunit 2; COPS2, COP9 signalosome complex subunit 2; CRP, C-reactive protein; CSG, chronic superficial gastritis associated with *H. pylori*; CTSF, cathepsin F; CXCL8, interleukin 8; CXCR2, C-X-C chemokine receptor type 2; DcR3, decoy receptor 3; DEK, protein DEK; DKK1, dickkopf-1 protein; ECLA, enhanced chemiluminescence assay; ECLIA, electro-chemiluminescence immunosorbent assay; ELISA, enzyme-linked immunosorbent assay; EphA1, erythropoietin-producing hepatocellular A1; ELISA, enzyme-linked immuno assay; ENOSF1, mitochondrial enolase superfamily member 1; EREG, proepiregulin; ERK, extra-cellular signal-regulated kinase; FGA, fibrinogen α chain; FGF-12, fibroblast growth factor 12; FIB, fibrinogen; FR-β, folate receptor β; G-17, gastrin-17; GC, gastric cancer; GHR, growth hormone receptor; GKN1, gastrokine 1; GPX3, glutathione peroxidase 3; GSN, gelsolin; IFNGR1, interferon gamma receptor 1; IGF-1, insulin-like growth factor 1; IHC, immunohistochemistry; IL-6, interleukin 6; IL-8, interleukin 8; IL-16, interleukin 16; IL-18, interleukin 18; IL-2R, Interleukin-2 receptor subunit α; ITGB6, integrin β-6; ITIH4, inter-α-trypsin inhibitor heavy chain H4; iTRAQ, isobaric tags for relative and absolute quantitation; LC, liquid chromatography; M0, non-metastatic; M+, metastatic; MB, magnetic bead-based; MS, mass spectrometry; MB-IMAC-Cu, magnetic beads-based immobilized metal-ion affinity chromatography; MRM, multiple reaction monitoring; Notch-3, neurogenic locus notch homolog protein 3; NT5E, ecto-5^E^-nucleotidase; OST, patients with other system malignant tumors; PD-1, programmed cell death protein 1; PDL-1, Programmed death-ligand 1; PEA, multiplex proximity extension assay; PGI, pepsinogen I; PGII, pepsinogen II; PGR, PGI to PGII ratio; ROC, receiver operating characteristic; S100A8, protein S100-A8; S100A9, protein S100A9; SAA, serum amyloid A; SAA1, serum amyloid A protein; sEGFR, soluble epidermal growth factor receptor; SELDI, surface-enhanced laser desorption/ionization; sIL-2R, soluble interleukin-2 receptor; SERPINA4, kallistatin; SHBG, sex hormone-binding globulin; SIRT6, sirtuin 6; sHLA-G, soluble human leukocyte antigen G; SLAMF8, signaling lymphocytic activation molecule family; SNCG, synuclein gamma protein; SOX3, transcription factor SOX-3; SRMS, tyrosine-protein kinase; TERF1, telomeric repeat binding factor 1; TGF-β1, transforming growth factor beta-3 proprotein; THB1, thrombospondin 1; TK1, thymidine kinase 1; TMT, tandem mass tags; TNFα, tumor necrosi factor α; TrxR, thioredoxin reductase; TSLP, thymic stromal lymphopoietin; TNFRSF19L, tumor necrosis factor receptor superfamily member 19L; VEGF, vascular endothelial growth factor; VN, vitronectin; WB, Western blot; WCX, weak cation exchange. ^(a)^ Protein name abbreviation is followed by the UniProtKB ID; ^(b)^ gene ontology biological process after interrogation with UniProtKB (https://www.uniprot.org/uniprotkb/; accessed on 30 September 2023): only the first main GO term is reported; ^(c)^ proteomics techniques used to analyze proteins are reported; ^(d)^ among the clinical characteristics detailed by authors, only those about tumor clinical stages (from I to V), histological types (intestinal, diffuse, mixed), gastric anatomic subsites (proximal, distal), pathological stages with the TNM system (pTNM stages, from T1 to T4), and presence of metastasis (M0, M1) are, if present, reported; ^(e)^ *nr*, number of individuals per group.

**Table 2 ijms-24-16931-t002:** List of non-blood circulating protein markers for GC diagnosis reported over the past 10 years.

Biomarker(s) ^(a)^	Proteomic Technology ^(b)^	Patients		Major Findings	Ref.
		Tumor Characteristics ^(c)^	Group (*nr*) ^(d)^		
** *Ascitic fluid* **					
PGC and POSTN	• LC-MS/MS • ELISA	• Clinical stage IV	GC (85)	In total, 299 differentially expressed proteins were quantified, 81 and 218 of which were up- and downregulated, respectively, in the ascitic fluids of GC. PGC and POSTN proteins distinguished malignant ascites from benign ones and were verified by ELISA, being thus potential candidate biomarkers of disease state.	[88]
** *Gastric juice* **					
SNCG (also from serum)	• ELISA	Not reported	GC (87), CPL (38), and C (44)	SNCG levels were higher in GC vs. CPL or C in both gastric juice and serum. The expression of SNCG in GJ and serum was significantly associated with tumor node metastasis stage, lymph node metastasis, tumor size, and drinking. The detection of SNCG in gastric juice and serum was reported as an ideal method of clinical diagnostic value for the early diagnosis of GC, with high specificity (90.91%) and sensitivity (83.91%) (positive predictive value = 94.81%; negative predictive value = 74.07%; 95% CI: 0.869–0.971; *p*< 0.0001).	[59]
Ela3A, PepA, GastL, Gastricsin, and CystD	• iTRAQ labeling • LC-MS/MS • WB	• Histology: 30% diffuse, 33% intestinal, 37% unknown/mix • invasion depth: 54% I–II, 46% III–IV	GC (70) and benign gastritis (17)	An increase in Ela3A together with a decrease in PepA, Gast, Gastricsin, and CystD occurred in gastric fluids of GC patients with high confidence. A three-biomarker panel of CystD + PepA + Ela3A was sufficient for initial GC diagnosis (sensitivity = 95.7%; specificity = 76.5%).	[89]
S100A9, GIF, and AAT	• 2DE • MS • WB	• Clinical stages: *discovery cohort*→33.3% I (early GC), 33.3% III 33.3% IV (late GC); *validation cohort*→51% early GC, 49% advanced GC	*Discovery cohort*: GC (9) and gastritis (3); *validation cohort*: GC (43) and gastritis (17)	Out of the 15 differential proteins identified, levels of S100A9, GIF, and AAT correlated with GC status. S100A9 and AAT (AUC = 0.81; *p* = 0.0013) were promising biomarker pairs for early GC diagnosis.	[90]
** *Saliva* **					
CSTB, TPI1, and DMBT1	• 1DE • 2D-DIGE • TMT • LC-MS/MS • ELISA	• Clinical stages: 75% I–II, 25% III–IV	*Discovery cohort*: GC (20) and C (20); *validation cohort*: GC (20) and C (20)	A total of 48 differential proteins were found (*p* < 0.05) between GC and C, including 7 up- and 41 downregulated proteins. Three proteins were successfully validated (CSTB, TPI1, and DMBT1). These proteins differentiated GC (*p* < 0.05) and, combined, showed a sensitivity of 85% and a specificity of 80% in GC detection with an accuracy of 0.93.	[91]
** *Urine* **					
SORT1 and VTN	• TMT • LC-MS/MS	• Grade 2 or grade 3 adenocarcinoma	*Discovery cohort*: GC (5) and C (5); *validation cohort*: GC (19) and C (12)	A total of 246 proteins were differentially expressed in GC cases. Some proteins more abundant in GC vs. C are already known to play crucial roles in GC progression (ephrin A1, pepsinogen A3, sortilin 1, SORT1, and VTN). Others had not previously been linked to GC (shisa family member 5, mucin-like 1, and leukocyte cell-derived chemotaxin 2). The overexpression of SORT1 and VTN in GC urines was confirmed in an independent set of urine samples.	[92]
ANXA11, CDC42, NAPA, and SLC25A4	• LC-MS/MS	Not detailed	*Discovery cohort*: GC (14) and patients with gastric lesions (109; SG, CAG, IM, or LGIN); *validation cohort*: GC (18) and patients with gastric lesions (114; SG, CAG, IM, or LGIN)	Urinary levels of ANXA11, CDC42, NAPA, and SLC25A4 were positively associated with gastric lesion progression; they may potentially predict the progression of gastric lesions and risk of GC occurrence.	[93]
ADAM12 and TFF1	• LC-MS/MS	Not detailed	*discovery cohort:* 18 patients; *training cohort:* 176 patients; *validation cohort:* 88 patients.	A urinary biomarker panel combining TFF1, ADAM12, and *H. pylori* significantly discriminated early GC vs. C in both training and validation cohorts.	[94]
ADAM12 and MMP-9/NGAL complex	• Protein array analysis • substrate gel electrophoresis • ELISA	• Histology: 23% diffuse, 77% intestinal • clinical stages: 75% I–II, 25% III–IV • invasion depth: 69% T1–T2, 31% T3–T4	GC (35) and C (35)	Urinary levels of the MMP-9/NGAL complex and ADAM12 were higher in GC vs. C (*p* < 0.001). Both the MMP-9/NGAL complex and ADAM12 were significant, independent diagnostic biomarkers for GC by multivariate analysis and distinguished between GC and C samples (AUC = 0.825, *p* < 0.001) in an ROC analysis.	[95]
EL	• WB	• Histology: 43% intestinal, 57% diffuse • clinical stages: 33% I–II, 67% III–IV • invasion depth: 33% T1–T2, 67% T3–T4 • degree of differentiation: 48% high or moderate, 52% poor or undifferentiated	GC (90) and C (57)	The EL content decreased by a ~9.9-fold average in GC vs. C (*p* < 0.0001), achieving a 0.967 AUC value for the ROC curve, demonstrating its high accuracy as a promising diagnostic marker for GC.	[96]

Abbreviations: 1-DE, one-dimensional electrophoresis; 2DE, two-dimensional electrophoresis; AAT, α-1-antitrypsin; ADAM12, disintegrin and metalloproteinase domain-containing protein 12; ANXA11, annexin A11; CAG, chronic atrophic gastritis; CDC42, cell division control protein 42 homolog; CPL, gastric precancerous lesions; CSTB, cystatin B; CystD, cystatin D; Da, Dalton; DMBT1, deleted in malignant brain tumors 1 protein; EL, endothelial lipase; Ela3A, elastase 3A; IM, intestinal metaplasia; MMP-9, matrix metalloproteinases-9; GastL, gastric lipase; GIF, gastric intrinsic factor; LGIN, low-grade intraepithelial neoplasia; MALDI-TOF, matrix-assisted laser desorption ionization time-of-flight; MMP9, matrix metallopeptidase 9; MS, mass spectrometry; NAPA, NSF attachment protein α; NGAL, neutrophil gelatinase-associated lipolalin; PepA, pepsin A; PGC, progastriscin; POSTN, periostin; ROC, receiver operating characteristic; S100A9, S100 calcium-binding protein A9; SLC25A4, solute carrier family 25 member 4; SG, superficial gastritis; SNCG, synuclein-gamma; SORT1, sortilin 1; TFF1, trefoil factor 1; TMT, tandem mass tags; TPI1, triosephosphate isomerase; VTN, vitronectin. ^(a)^ Protein name abbreviation is followed by the UniProtKB ID; ^(b)^ proteomics techniques used to analyze proteins are reported; ^(c)^ among the clinical characteristics detailed by authors, only those about tumor clinical stages (from I to V), histological types (intestinal, diffuse, mixed), gastric anatomic subsites (proximal, distal), pathological stages with the TNM system (pTNM stages, from T1 to T4), and presence of metastasis (M0, M1) are, if present, reported; ^(d)^ *nr*, number of individuals per group.

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
