# Peer review of "Circulating Proteins as Diagnostic Markers in Gastric Cancer"

_ijms, 2023, doi:10.3390/ijms242316931_

Round 1

Reviewer 1 Report

Comments and Suggestions for Authors

Reviewer comments

The review article on “Circulating proteins as diagnostic markers in gastric cancer” has discussed the circulating protein biomarkers in GC and details their application as potential biomarkers for GC diagnosis. The article is well-planned and discussed very well. However, there are a few suggestions to incorporate in the revised file.

1.      Discuss the future perspective of protein biomarkers.

2.      Discuss the limitations of protein markers and how to overcome them.

3.      According to your review work give a concluding remark on prominent protein biomarkers for GC and among all listed techniques which are most efficient for early and sensitive detection.

4.      Check the reference format (year is in bold font) and change all accordingly.

Reviewer 2 Report

Comments and Suggestions for Authors

In my opinion, the article contains little analysis and systematization of data.

1. So, what methods are most often used to analyze circulating markers (their advantages/disadvantages, etc.)?

2. what markers are repeated in studies by various authors?

3. Do data from different authors on the same indicators correlate with each other?

4. What markers are associated with tumor characteristics and provide information about the heterogeneous nature of gastric cancer?

5. Which multivariate methods are most often used?

6. What are the prospects for introducing methods into clinical practice?

Round 2

Reviewer 2 Report

Comments and Suggestions for Authors

The authors made changes to the manuscript in accordance with the comments of the reviewers. I believe that in its present form the manuscript can be recommended for publication.